# Abdominal fat depots and their association with insulin resistance in patients with type 2 diabetes

Umesh Kumar Garg[1], Nitish Mathur[1], Rahul Sahlot[1], Pradeep Tiwari[1,2], Balram Sharma[1], Aditya Saxena[3], Raj Kamal Jainaw[4], Laxman Agarwal[4], Shalu Gupta[4], Sandeep Kumar Mathur[1] *

1 Department of Endocrinology, Sawai Man Singh Medical College and Hospital, Jaipur, India, 2 Department of Chemistry, School of Basic Sciences, Manipal University Jaipur, Jaipur, India, 3 Department of Computer Engineering & Applications, GLA University, Mathura, India, 4 Department of Surgery, Sawai Man Singh Medical College and Hospital, Jaipur, India

* drsandeepmathur@rediffmail.com

## Abstract

### Background

Asian-Indians show thin fat phenotype, characterized by predominantly central deposition of excess fat. The roles of abdominal subcutaneous fat (SAT), intra-peritoneal adipose tissue, and fat depots surrounding the vital organs (IPAT-SV) and liver fat in insulin resistance (IR), type-2 diabetes (T2D) and metabolic syndrome (MetS) in this population are sparsely investigated.

### Aims and objectives

Assessment of liver fat, SAT and IPAT-SV by MRI in subjects with T2D and MetS; and to investigate its correlation with IR, specifically according to different quartiles of HOMA-IR.

### Methods

Eighty T2D and the equal number of age sex-matched normal glucose tolerant controls participated in this study. Abdominal SAT, IPAT-SV and liver fat were measured using MRI. IR was estimated by the Homeostatic Model Assessment for Insulin Resistance (HOMA-IR).

### Results

T2D and MetS subjects have higher quantity liver fat and IPAT-SV fat than controls ($P = 9 \times 10^{-4}$ and $4 \times 10^{-4}$ for T2D and $10^{-4}$ and $9 \times 10^{-3}$ for MetS subjects respectively). MetS subjects also have higher SAT fat mass ($P = 0.012$), but not the BMI adjusted SAT fat mass ($P = 0.48$). Higher quartiles of HOMA-IR were associated with higher BMI, $W:H$ ratio, waist circumference, and higher liver fat mass (ANOVA Test $P = 0.020, 0.030, 2 \times 10^{-6}$ and $3 \times 10^{-3}$ respectively with F-values 3.35, 3.04, 8.82, 4.47 respectively). In T2D and MetS subjects, HOMA-IR showed a moderately strong correlation with liver fat ($r = 0.467, P < 3 \times 10^{-5}$ and $r = 0.493, P < 10^{-7}$), but not with SAT fat and IPAT-SV. However, in MetS subjects IPAT-SV

**Data Availability Statement:** All relevant data are within the paper and its Supporting Information files.

**Funding:** Dr sandeep kumar mathur SKM) received grant from MMRS (Metabolic and molecular research society Jaipur), ICMR (Indian council of medical research) and RSSDI (Research society for the study of diabetes in india) ICMR grant no. 5/4/2012-RMC RSSDI grant no. RSSDI/HQ/Grants/2018/4647 Metabolic and molecular research society Jaipur, Indian council of medical research, Research society for the study of diabetes in india https://www.icmr.gov.in/ https://www.rssdi.in/newwebsite The funders had no role in study design, data collection and analysis, decision to publish, or preparation of the manuscript.

**Competing interests:** The authors have declared that no competing interests exist.

fat mass showed borderline correlation with IR ($r = 0.241$, $P < 0.05$), but not with the BMI adjusted IPAT-SV fat mass ($r = 0.13$, $P = 0.26$). In non-T2D and non-MetS subjects, no such correlation was seen. On analyzing the correlation between the three abdominal adipose compartment fat masses and IR according to its severity, the correlation with liver fat mass becomes stronger with increasing quartiles of HOMA-IR, and the strongest correlation is seen in the highest quartile ($r = 0.59$, $P < 10^{-3}$). On the other hand, SAT fat mass tended to show an inverse relation with IR with borderline negative correlation in the highest quartile ($r = -0.284$, $P < 0.05$). IPAT-SV fat mass did not show any statistically significant correlation with HOMA-IR, but in the highest quartile it showed borderline, but statistically insignificant positive correlation ($P = 0.07$).

## Conclusion

In individuals suffering from T2D and MetS, IR shows a trend towards positive and border-line negative correlation with liver fat and SAT fat masses respectively. The positive trend with liver fat tends to become stronger with increasing quartile of IR. Therefore, these findings support the theory that possibly exhaustion of protective compartment's capacity to store excess fat results in its pathological deposition in liver as ectopic fat.

## Introduction

Insulin resistance (IR) is defined as a suboptimal biological response or inefficient action of insulin in its responsive organs and tissues. IR clusters with abdominal obesity and has been implicated in the pathophysiology of type-2 diabetes (T2D) and metabolic syndrome (MetS) [1, 2]. There could be a complex inter-relationship between abdominal fat deposition, IR and the pathogenic pathways of T2D and MetS. However, the cut-off values of these quantitative traits of abdominal girth for MetS have been a subject of debate and are generally defined arbitrarily by different professional organizations. However, racial differences in the quantitative trait of adiposity and consequent dysmetabolism are well known [3–5]. Some races like Asian-Indians are not only at high risk for T2D and MetS, but they also exhibit a unique phenotype, so-called the 'Thin fat phenotype' [6, 7]. This phenotype is characterized by higher body fat content which is predominantly deposited in the central and visceral compartments. For a given level of Body mass index (BMI), they are more insulin resistant and have a higher prevalence of T2D and MetS [8–12].

Measurement of intra-abdominal visceral fat accumulation is an important step for the assessment of adiposity and prediction of T2D and MetS risk. The distribution of abdominal fat as subcutaneous adipose tissue, adipose depots surrounding the intraperitoneal organs like liver, pancreas, kidneys and intestine etc., i.e., non-ectopic visceral and fat deposited in the liver and other vital organ (in terms of pathophysiology, so called the ectopic fat), is a dimension of obesity that cannot be adequately assessed by anthropometric measurements like BMI, Waist Circumference and Waist: Hip Ratio (W: H). Computer Tomography (CT) or Medical Resonance (MR) imaging techniques are currently considered to be the methods of choice for the assessment of abdominal fat distribution in these compartments [13].

Several previous studies have explored the relationship between body fat-distribution and its association with T2D, MetS, and IR. Raji *et. al* have shown that at the same BMI, Asian-Indian healthy subjects have higher IR and significantly greater total abdominal fat and

visceral fat masses than Caucasians [8] and in all subjects, measures of fat mass correlate with IR. On the contrary Banerji *et al.*, in their study of healthy non-obese Asian Indian men, found them to be insulin resistant with a high percentage of body fat relative to their BMI and muscle mass but their IR correlated with visceral adipose tissue fat mass but not with the subcutaneous fat mass [9]. Anjana *et al* from India have shown in their study that Visceral and central abdominal fat showed a strong association with type 2 diabetes. Both measures correlated well with each other [14]. Mishra and Vikram reported that higher body fat especially truncal and abdominal fat at lower BMI in Asian-Indians [15]. Ramachandran and his group also found visceral adipose tissue fat mass to be associated with insulin secretion in males [16]. But to the best of our knowledge, the relationship between abdominal fat distributions in SAT, IPAT-SV and liver fat compartments, particularly in the context of the severity of IR remain unexplored in this population.

Therefore, the objective of this study is to find the relationship between MRI assessed abdominal fat distribution in SAT, IPAT-SV, liver fat and IR. The other objective was to assess the relationship in different quartiles of IR in these subjects. The secondary objective was to find an association between abdominal fat compartmentalization *viz.* SAT, IPAT-SV and liver fat with MetS.

## Methodology

### Study subjects

Our study is an analytical cross-sectional study conducted from 2019 to 2021 at a tertiary level hospital in northwest India. One hundred and sixty individuals, comprising eighty non-T2D controls (Male: Female = 40:40) and eighty people with T2D (Male: Female = 41:39) were recruited for this study after obtaining their written consent. The institutional ethics committee of SMS Medical College, Jaipur approved the study protocol. Participants' inclusion criteria were T2D diagnosed as per American Diabetes Association (ADA) (2018) standards and normal individuals without T2D in another arm (confirmed by Oral Glucose Tolerance Test -OGTT). The exclusion criteria were the presence of infection, malignancy, and drugs affecting body fat such as thiazolidinediones and glucocorticoids.

### Clinical & biochemical assessment

The anthropometric measurements, including body weight, height, waist circumference, and hip circumference were measured using standard methods in all participants. Waist to hip (W: H) ratio, and BMI (in kg/m$^2$) were then calculated. Supine blood pressure was measured using a mercury sphygmomanometer (BPMR-120 Diamond deluxe, Industrial electronic and allied products, Maharashtra, India) after 10 minutes of rest. Blood samples were obtained at 8:00 am after an overnight fast for at least 8 hours. Oral Glucose Tolerance Test (OGTT) was carried out with 75 gm. anhydrous glucose in subjects without prior history of T2D to confirm a normal glucose tolerant state. Various biochemical parameters (e.g., Plasma glucose (FPG), serum total cholesterol, triglycerides, low-density lipoprotein cholesterol (LDL-C), high-density lipoprotein cholesterols (HDL-C), and very-low-density lipoprotein cholesterol (VLDL-C)) were measured on Kopran AU/400 (Olympus Corporation, Shinjuku, Tokyo, Japan) fully automated analyzer. Serum insulin was measured using a chemiluminescent immunometric assay (IMMULITE 2000 machine, Siemens, Germany). HbA1c was measured by the HPLC method on TOSOH automated analyzer. HOMA-IR was calculated using the Homeostasis model assessment (HOMA IR) formula: fasting insulin (μIU/ml) x fasting plasma glucose (mmol/l) / 22.5, as described by Matthews *et. al.* [17] previously. We further log- transformed HOMA-IR values to conform to normality.

## Determination of metabolic syndrome parameter

According to the National Cholesterol Education Program (NCEP) Adult Treatment Panel-III (ATP-III) definition, metabolic syndrome is present if three or more of the following five criteria are met: 1) increased waist circumference ($\geq$90 cms for men, $\geq$80 cms for women); 2) elevated triglycerides ($\geq$150 mg/dl); 3) low HDL cholesterol ($<$40 mg/dl in men, $<$50 mg/dl in women); 4) hypertension ($\geq$130/$\geq$85 mmHg); and 5) fasting glucose ($\geq$100 mg/dl). In addition, existing drug treatment for dyslipidaemia, dysglycemia, raised blood pressure would also be qualifying criteria.

There were 67 subjects with MetS and 91 were without MetS (2 subjects unclassified due to missing lipid profile data).

## Radiological investigations

Abdominal fat content and distribution among abdominal SAT, IPAT-SV fat mass and liver fat were estimated by MRI. MRI scan procedure: The MRI scans were done at the S.M.S, hospital in the department of radiology, using 3 tesla Philips Ingenia Machine. The single investigator who interpreted the scans on Osirix software was unaware of the clinical status of the study subjects. A single scan (3 mm) of the abdomen was done at the level of L4-L5 vertebrae and analyzed for a cross-sectional area of adipose tissue, which was expressed in centimeters squared. The parameters studied included liver fat, IPAT-SV fat mass and abdominal SAT fat mass. IPAT-SV fat mass, which represents the adipose tissue depots surrounding the intra-peritoneal organs like liver, pancreas, kidneys and intestine etc. (including the omentum, perinephric adipose tissue etc, but excluded the fat contained in these organs as well as intra-muscular fat) was distinguished from abdominal SAT fat by tracing along the fascial plane defining the internal abdominal wall and the area was calculated in centimeters square. Liver fat was measured using liver intensity on Osirix software using Dixon method (Liver Fat percentage = 100X ((Signal intensity liver/signal intensity spleen) on in phase T1- (signal intensity liver/signal intensity spleen) on out phase T1)/2x (signal intensity liver/signal intensity spleen) on in phase T1).

Statistical analysis was done on SPSS statistics 26.0, using the student T-test for parametric variables. while Mann Whitney U test was applied for the non-parametric test. A $P$-value $< 0.05$ was considered significant. The correlation coefficient ($r$) was calculated using Pearson's formula.

# Results

## (a). General characteristics

The general characteristics of the study subjects are shown in Table 1. A total of 160 subjects were studied. The male to female ratio was 81:79. There were an equal number of subjects with T2D and without T2D (80 in each). Both the groups were matched for age and sex. Subjects with T2D had higher Weight ($P = 0.0037$), BMI ($P = 3 \times 10^{-3}$), Waist circumference ($P < 0.05$), and WHR ($P = 0.02$). This group also had higher triglyceride levels ($P = 0.02$), HOMA-IR ($P < 5 \times 10^{-3}$), FBS ($P < 5 \times 10^{-3}$), HbA1c ($P < 5 \times 10^{-3}$). On comparison of subjects with and without MetS, total 158 subjects were included in the final analysis as two subjects were unclassifiable due to missing data. There were 67 subjects with MetS, and 91 subjects didn't have MetS. The former group had higher Weight ($P = 10^{-3}$), BMI ($P < 10^{-3}$), Waist ($P < 10^{-3}$), WHR ($P = 3 \times 10^{-3}$), Triglycerides ($P < 10^{-3}$), LDL-C ($P = 0.04$), HOMA-IR ($P < 5 \times 10^{-3}$), FPG ($P < 10^{-3}$), HbA1c ($P < 10^{-3}$) and lower HDL-C ($P = 0.04$). Due to high standard deviation value (SD); log transformation of HOMA-IR was carried out and included in analysis. A heat map of correlations across traits of interest is presented in Fig 1.

**Table 1. General Characteristics of the study subjects and observations.**

| | T2D | Non-T2D | *P*–value (T2D vs Non-T2D) | With MetS | Without MetS | *P*-value (MS vs Without MS) |
|---|---|---|---|---|---|---|
| Study population (n) | 80/160 | 80/160 | | 67/158 | 91/158 | |
| Age (Years) | 57.83±10.65 | 54.38±11.74 | 0.055 | 56.91 ± 11.01 | 55.66 ± 11.64 | 0.495 |
| Height (in Mt) | 1.61±0.09 | 1.61±0.09 | 0.804 | 1.61 ± 0.076 | 1.62 ± 0.09 | 0.442 |
| Weight (Kg) | 64.52±10.36 | 59.59±10.62 | 0.003 | 65.78 ± 9.49 | 59.35±10.90 | 0.0001 |
| BMI (Kg/m$^2$) | 24.79±4.35 | 22.89±3.62 | 0.003 | 25.52 ± 4.13 | 22.62±3.66 | < 0.001 |
| Waist (cm) | 93.73±10.10 | 87.15±10.08 | <0.005 | 97.31 ± 7.98 | 85.26±9.36 | < 0.001 |
| WHR | 0.97±0.07 | 0.94±0.07 | 0.026 | 0.98 ± 0.07 | 0.94 ± 0.07 | 0.003 |
| Total Cholesterol (mg/dl) | 176.66±51.20 | 165.39±43.64 | 0.144 | 182.2 ± 51.51 | 163.21 ± 43.24 | 0.017 |
| Triglycerides (mg/dl) | 160.18±74.64 | 133.47±72.86 | 0.026 | 173.32 ± 71.74 | 126.87 ± 71.49 | 0.0001 |
| HDL-C (mg/dl) | 44.47. ±10.98 | 44.54±7.65 | 0.965 | 42.8 ± 9.79 | 45.92 ± 8.99 | 0.047 |
| LDL-C (mg/dl) | 96.57±34.69 | 88.77±33.72 | 0.162 | 99.398 ± 35.51 | 88.199 ± 32.71 | 0.048 |
| Log HOMA-IR | 0.65±0.36 | 0.12±0.25 | < 0.005 | 0.637±0.385 | 0.198±0.323 | <0.001 |
| Fasting plasma glucose (mg/dl) | 173.60 ±68.75 | 87.33±12.38 | < 0.005 | 169.69 ± 73.22 | 102.27 ± 40.75 | < 0.001 |
| HbA1c (%) | 8.21±1.74 | 5.25±0.75 | < 0.005 | 7.99 ± 1.78 | 5.84 ± 1.64 | < 0.001 |

## (b). MRI parameters of abdominal adipose tissue mass

Subjects with T2D had higher liver fat ($P = 10^{-3}$, $P_{BMI-adjusted} = 0.03$) and IPAT-SV fat mass ($P = 10^{-3}$, $P_{BMI-adjusted} = 0.04$) as compared to non- T2D subjects, but both the groups had similar abdominal SAT fat mass ($P = 0.28$, $P_{BMI-adjusted} = 0.92$) (Table 2).

The MetS subjects have higher fat mass in all three abdominal adipose tissue compartments studied, *i.e.*, liver fat ($P = 10^{-4}$, $P_{BMI-adjusted} = 0.03$) and IPAT-SV ($P = 0.018$, $P_{BMI-adjusted} = 10^{-4}$) and abdominal SAT ($P = 0.01$) depots.

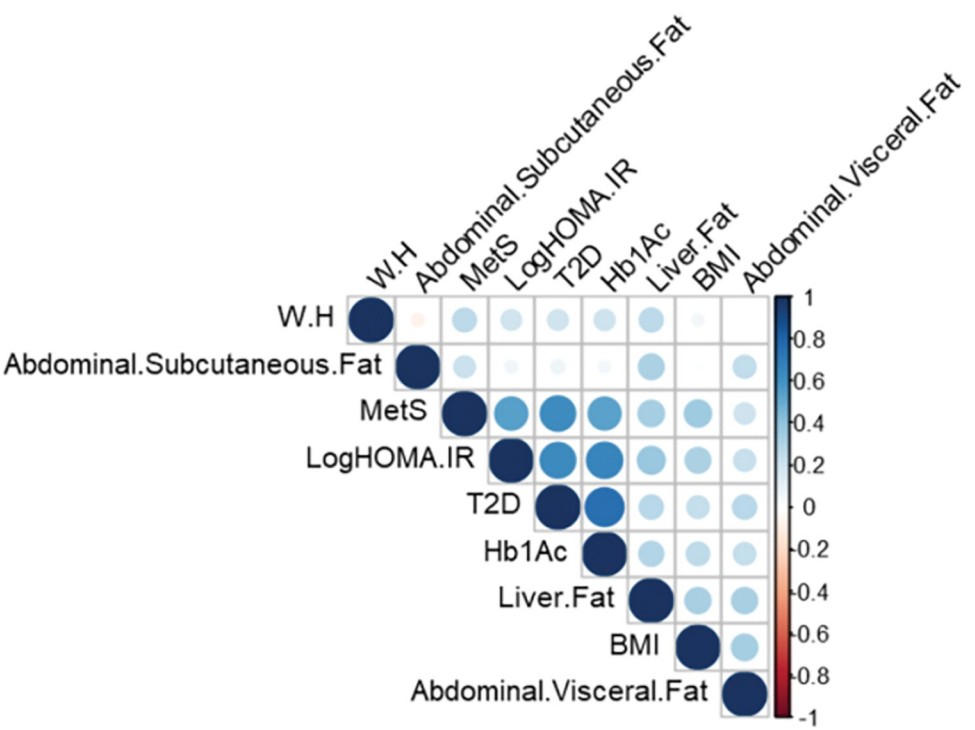

**Fig 1. Correlation plot across various traits of interest.**

**Table 2. Comparison MRI assessed adipose tissue mass in abdominal subcutaneous fat (SAT), intra-peritoneal adipose tissue, and fat depots surrounding the vital organs (IPAT-SV) and ectopic liver compartments between T2D and NGT as well as MetS and Non-MetS groups.**

| | T2D | Non- T2D | *P*-value<br>(T2D vs ND) | MetS | Without MetS | *P* -value<br>(MetS vs Non- MetS) |
|---|---|---|---|---|---|---|
| IPAT-SV fat (cm$^2$) | 162.28±58.79 | 130.95±54.12 | **0.001** | 159.99±64.49 | 136.34±51.62 | **0.018** |
| BMI adjusted IPAT-SV fat mass (cm$^2$) | 6.58±2.26 | 5.77±2.53 | **0.043** | 6.27±2.45 | 6.10±2.42 | **0.0001** |
| SAT fat mass(cm$^2$) | 150.21±63.58 | 139.12±61.47 | 0.282 | 160.24±73.08 | 132.99±50.84 | **0.012** |
| BMI adjusted SAT fat mass (cm$^2$) | 6.16±2.73 | 6.21±3.32 | 0.924 | 6.39±3.14 | 6.03±2.97 | 0.48 |
| Liver fat (%) | 8.56±3.93 | 6.57±3.10 | **0.001** | 8.98±4.20 | 6.52±2.82 | **0.0001** |
| BMI adjusted liver fat (%) | 0.35±0.15 | 0.29±0.16 | **0.035** | 0.35±0.17 | 0.29±0.15 | **0.03** |

### (c). Relationship between adipose tissue compartment fat and IR

The relationship between three abdominal adipose compartment fat mass and IR is shown in Table 3. In subjects with T2D, there was a strong correlation between liver fat and IR (r = 0.609, *P* < 0.05). But it was not observed in non-T2D subjects. In the T2D as well as non-T2D subjects, no significant correlation between abdominal SAT or IPAT-SV fat masses with IR was noticed.

### (d). Relationship between adipose tissue compartment fat mass and IR in subjects with MetS

In subjects with MetS there was a very strong correlation between HOMA-IR and liver fat (r = 0.493, $P < 10^{-7}$) (Table 3). There was a trend toward positive correlation of log transformed HOMA-IR with IPAT-SV fat mass, but it was not statistically significant (*r* = 0.24, *P* >0.05). No such correlation was observed in subjects without MetS.

### (e). Abdominal adipose tissue fat mass and its correlation with HOMA-IR and other parameters according to the severity of IR

In the comparison of subjects according to IR, higher quartiles were associated with an increased amount of liver fat (F = 4.776, *P* = 0.0033 in ANOVA one-way test) (Table 4) and borderline significant higher amount of IPAT-SV fat mass (F = 2.527, *P* = 0.05). No difference in subcutaneous fat mass was observed with increasing IR. Waist circumference, Hip circumference, WHR, BMI and serum triglyceride levels were also higher in the higher quartiles ($P = 2 \times 10^{-6}$, $9 \times 10^{-3}$, 0.03, 0.02, 0.04 respectively and F = 8.83, 3.93, 3.044, 3.35, 2.79 respectively).

There was a significant positive correlation between the liver fat mass and log HOMA-IR. This correlation became stronger in higher quartiles when compared to lower quartiles (Table 5 (*i.e.*, *r* = 0.002, 0.139, 0.349, 0.590, in quartiles I, II, III, and IV respectively). On the

**Table 3. Correlation of insulin resistance (HOMA-IR) to the fat mass in various adipose depots.**

| Correlation coefficient of HOMA-IR to fat depots | T2D | Non-T2D | With MetS | Without MetS |
|---|---|---|---|---|
| Correlation of Log HOMA-R to adjusted IPAT-SV fat mass | 0.162 (*P* = 0.170) | -0.100 (*P* = 0.388) | 0.241 (*P* = 0.056) | -0.016 (*P* = 0.877) |
| Correlation of Log HOMA-R to BMI adjusted IPAT-SV fat mass (cm$^2$) | -0.004 (*P* = 0.970) | -0.146 (*P* = 0.210) | 0.137 (*P* = 0.260) | -0.049 (*P* = 0.652) |
| Correlation of Log HOMA-R to SAT fat mass | -0.020 (*P* = 0.865) | 0.042 (*P* = 0.714) | -0.119 (*P* = 0.200) | -0.002 (*P* = 0.978) |
| Correlation of Log HOMA-R to BMI adjusted SAT fat mass (cm$^2$) | -0.123 (*P* = 0.300) | -0.006 (*P* = 0.954) | -0.194 (*P* = 0.150) | -0.033 (*P* = 0.763) |
| Correlation of Log HOMA-R to liver Fat | 0.467 (*P* = 0.00003) | -0.107 (*P* = 0.362) | 0.493 (***P* = 0.0000001**) | -0.063 (*P* = 0.566) |
| Correlation of Log HOMA-R to BMI adjusted liver fat (%) | 0.359 (***P* = 0.002**) | -0.140 (*P* = 0.228) | 0.383 (***P* = 0.002**) | -0.063 (*P* = 0.339) |

**Table 4. Quartile (of HOMA-IR) wise characteristics and analysis.**

| Group and parameters | Quartile 1 (Mean ± SD) | Quartile 2 (Mean ± SD) | Quartile 3 (Mean ± SD) | Quartile 4 (Mean ± SD) | ANOVA Test (p & F values) |
|---|---|---|---|---|---|
| Age (Years) | 54.75±13.11 | 56.65±10.49 | 57.57±9.90 | 55.47±11.41 | *P* = 0.698 F = 0.477 |
| Height (in Mt) | 1.61±0.09 | 1.61±0.09 | 1.63±0.10 | 1.61±0.73 | *P* = 0.655 F = 0.539 |
| Weight (Kg) | 59.13±12.62 | 59.77±8.83 | 64.65±10.12 | 64.67±9.87 | *P* = 0.024 F = 3.220 |
| BMI (Kg/m$^2$) | 22.69±4.25 | 23.02±2.85 | 24.47±4.23 | 25.17±4.41 | *P* = 0.020 F = 3.351 |
| Waist circumference (cm) | 85.67±10.05 | 88.44±10.40 | 91.03±10.0 | 96.71±8.65 | *P* = 0.000002 F = 8.829 |
| Hip circumference (cm) | 91.18±10.36 | 93.17±9.68 | 95.74±8.52 | 98.07±8.97 | *P* = 0.009 F = 3.925 |
| WHR | 0.94±0.07 | 0.95±0.06 | 0.95±0.08 | 0.98±0.07 | *P* = 0.030 F = 3.043 |
| Total Cholesterol (mg/dl) | 173.60±45.75 | 163.63±41.57 | 163.86±51.11 | 185.53±50.22 | *P* = 0.210 F = 1.523 |
| Triglycerides (mg/dl) | 119.0±55.95 | 148.53±83.21 | 154.35±88.46 | 165.89±58.18 | *P* = 0.042 F = 2.792 |
| HDL-C (mg/dl) | 46.24±9.55 | 43.67±7.85 | 46.33±11.33 | 41.94±8.24 | *P* = 0.122 F = 1.957 |
| LDL-C (mg/dl) | 99.08±33.01 | 85.03±33.06 | 81.74±31.51 | 104.81±34.59 | *P* = 0.007 F = 4.172 |
| Log HOMA-IR | -0.08±0.19 | 0.22±0.07 | 0.47±0.07 | 0.93±0.25 | *P* = <0.001 F = 23.780 |
| Fasting plasma glucose(mg/dl) | 84.53±11.97 | 93.51±15.70 | 138.91±38.70 | 204.89±78.93 | *P* = <0.001 F = 58.170 |
| HbA1c (%) | 5.15±0.68 | 5.85±1.58 | 7.35±1.69 | 8.62±1.71 | *P* = <0.001 F = 42.510 |
| IPAT-SV fat mass (cm$^2$) | 134.80±55.53 | 134.77±56.35 | 157.96±37.64 | 163.13±71.52 | *P* = 0.059 F = 2.526 |
| SAT fat mass (cm$^2$) | 128.55±61.51 | 152.16±57.62 | 143.99±45.85 | 153.78±78.58 | *P* = 0.29 F = 1.259 |
| Liver fat (%) | 6.96±3.70 | 6.38±2.13 | 7.75±2.79 | 9.36±4.89 | *P* = 0.003 F = 4.776 |

other hand, the correlation of IPAT-SV fat mass with HOMA-IR shows positive trends with each increasing quartile ($r = 0.18$, $P = 0.07$ in quartile IV), but it was not statistically significant. Abdominal SAT fat mass showed almost statistically significant negative correlation with log HOMA-IR in the highest quartile of IR ($r = -0.28$ $P \approx 0.05$ in quartile IV).

## Discussion

It was observed in this study that T2D and MetS subjects have significantly higher liver fat mass, which also tended to show a significant positive correlation with IR. Moreover, this relationship became stronger with increasing quartiles of HOMA-IR. Such a relationship was not seen in non-T2D or non- MetS individuals. Though IPAT-SV fat and abdominal SAT fat mass did not show significant association with IR, but in its highest quartile, HOMA-IR tended to show a borderline negative correlation with abdominal SAT fat mass. In other words, these findings suggest that IR's relationship with abdominal adipose compartments fat distribution varies with its severity and it is different in dysmetabolic conditions like T2D and MetS. As the increasing quartiles of HOMA-IR were also associated with higher BMI, waist circumference and W:H ratio, these findings provide indirect evidence that in subjects with a tendency towards predominantly central obesity, under the situation of positive calorie balance, the liver fat and possibly abdominal SAT fat masses respectively play causative and protective roles in the pathogenesis of IR. These findings are also consistent with the nutrient overflow theory of adipose tissue redistribution in Asian-Indians. According to this theory, Asian-Indians, who

**Table 5. Correlation of insulin resistance (HOMA-IR) to various fat depots according to the severity of insulin resistance (Quartile wise).**

| Correlation coefficient of HOMA-IR to fat depots | Quartile 1 | Quartile 2 | Quartile 3 | Quartile 4 |
|---|---|---|---|---|
| Correlation of Log HOMA-R to IPAT-SV fat mass | -0.216 (*P* = 0.143) | 0.131 (*P* = 0.110) | -0.1715 (*P* = 0.121) | 0.185 (*P* = 0.078) |
| Correlation of Log HOMA-R to **SAT fat mass** | -0.019 (*P* = 0.440) | 0.110 (*P* = 0.101) | 0.0129 (*P* = 0.210) | **-0.2843 (*P* = 0.05)** |
| Correlation of Log HOMA-R to liver Fat | 0.002 (*P* = 0.413) | 0.139 (*P* = 0.090) | **0.349 (*P* = 0.021)** | **0.590 (*P* = 0.001)** |

show thin fat phenotype, under the condition of positive caloric balance, the ability of their abdominal SAT fat compartment to store excessive fat (hence its protective role) is well exhausted, which leads to ectopic fat deposition in other organs like liver, muscle etc. This fat plays a major role in the pathogenesis of IR, T2D and MetS [18]. The findings of the present study add to this theory that the maladaptive pathophysiologic adipose tissue redistribution of fat is a characteristic feature of the subjects in the highest quartile of HOMA-IR. Most of these subjects also manifest clinically with T2D and MetS. However, it is worth mentioning here that in the highest quartile of IR, abdominal SAT fat only showed a borderline negative correlation with HOMA-IR.

Several previous studies have investigated the body fat content and distribution in Asian Indians and its association with IR, T2D and MetS [6–9, 14–16, 19–23]. Some of them also compared their adiposity parameters with those of other races like Caucasians and African Americans [6–9]. Most of them find that Asian Indians not only had excess visceral adipose tissue fat mass but also shows an association with IR, T2D and MetS. It is generally believed that higher T2D and MetS risk in Asian Indians is because of their higher visceral adipose tissue fat content and this pattern of fat deposition is known as the "thin fat" phenotype. There are several limitations of these studies. For example, they were from different parts of India, they had a relatively smaller sample size and used different methods for fat estimation (CT scan, MRI, ultrasonography or DEXA) etc. Only a few studies had investigated the relationship between IR and fat mass in three abdominal adipose tissue compartments (i.e., abdominal SAT fat, IPAT-SV fat and liver & other vital organ fat) separately [22, 24]. Also, to the best of our knowledge, no study has explored the relationship between fat mass in these adipose compartments and IR according to different degrees of severity of HOMA-IR. A recently published study by Chevli PA *et. al*, has found a significant correlation between ectopic liver fat with IR (HOMA-IR) in people of different ethnicity. They found heterogenicity in various ethnicity and various fat depots [24]. In this study, they have reported Pearson correlation of HOMA-IR with fat content in different adipose compartments as follows: r- value of 0.29 with visceral adipose tissue, 0.09 with abdominal subcutaneous and -0.23 with hepatic fat, all correlation significant at a $P < 0.01$ [25]. The finding of negative correlation of IR with hepatic fat in this study is not consistent with those of our study and is difficult to explain. However, our results (r-value of 0.46 in T2D subjects and 0.49 in people with MetS) are consistent with the findings of a previous study by Yatsuya *et. al.* [26]. They reported a positive correlation between HOMA-IR and hepatic fat (R = 0.59). In another study, A G Rao *et. al.* reported that ectopic liver fat as measured by liver density on CT scan contributed to the prediction of IR, in addition to anthropometric measurement [22]. On the other hand, fat measured by MRI or DXA in their study doesn't add additional information to that provided by anthropometric measurement. Therefore, these findings suggest that ectopic liver fat is more informative than MRI measures of trunk or visceral fat for IR prediction, which is consistent with the findings of the present study.

It was observed in the present study that IR in its highest quartile show a strong positive and borderline negative correlation with liver fat and abdominal SAT fat mass respectively. Therefore, it points towards the role of fat redistribution from abdominal SAT to the liver fat compartment in the pathophysiology of high IR. Though T2D and MetS subjects in the present study had higher IPAT-SV fat mass, it shows only borderline association with IR. These findings are consistent with the fact that surgical removal of omentum either alone or in combination with bariatric surgery doesn't improve IR [27–29]. Additionally, there are several clinical, epidemiological and animal models' pieces of evidence that ectopic deposition of fat in the liver, muscle etc. plays an important role in the pathogenesis of IR, T2D and MetS [30–35]. Therefore, from point of view of understanding the core pathophysiologic mechanisms of IR,

T2D and MetS in Asian Indians, what seems to be more important is to explore molecular mechanisms of the fat deposition in the liver and possibly inability of abdominal SAT to expand and store fat under the condition of calorie excess, rather than studying IPAT-SV alone [36, 37].

There are several limitations of the present study that it had a cross-sectional design, it is a single center study and has relatively smaller sample size etc. Therefore, there is a need for large scale, multi-center, longitudinal or interventional studies to investigate the effect of change in caloric balance on fat distribution in three abdominal compartments and its relationship with insulin resistance and the occurrence of T2D and MS.

## Conclusion

The relationship between IR and quantity of fat in abdominal SAT, IPAT-SV and liver compartments varies from lower to higher quartiles of HOMA-IR. In the highest quartile, the liver fat shows a causative and abdominal SAT show a borderline protective relationship with IR. Though subjects with T2D and MetS had higher IPAT-SV fat mass, but it did not show significant correlation with HOMA- IR.

## Supporting information

**S1 File.**
(XLSX)

## Author Contributions

**Conceptualization:** Umesh Kumar Garg, Balram Sharma, Sandeep Kumar Mathur.

**Data curation:** Nitish Mathur, Pradeep Tiwari, Aditya Saxena.

**Formal analysis:** Umesh Kumar Garg, Nitish Mathur, Rahul Sahlot, Pradeep Tiwari, Aditya Saxena, Sandeep Kumar Mathur.

**Funding acquisition:** Sandeep Kumar Mathur.

**Investigation:** Nitish Mathur, Rahul Sahlot.

**Methodology:** Umesh Kumar Garg, Balram Sharma, Sandeep Kumar Mathur.

**Project administration:** Balram Sharma, Sandeep Kumar Mathur.

**Resources:** Umesh Kumar Garg, Raj Kamal Jainaw, Laxman Agarwal, Shalu Gupta.

**Supervision:** Umesh Kumar Garg, Nitish Mathur, Rahul Sahlot, Sandeep Kumar Mathur.

**Validation:** Umesh Kumar Garg, Sandeep Kumar Mathur.

**Writing – original draft:** Umesh Kumar Garg, Nitish Mathur, Rahul Sahlot, Balram Sharma, Sandeep Kumar Mathur.

**Writing – review & editing:** Umesh Kumar Garg, Sandeep Kumar Mathur.

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
