## [Decision Letter · Decision Letter 0]

6 Sep 2022

PONE-D-22-13467It is the ectopic liver fat, not the abdominal visceral fat that shows association with insulin resistance, Type-2 Diabetes and metabolic syndrome in Asian Indians.PLOS ONE

Dear Dr. Mathur,

Thank you for submitting your manuscript to PLOS ONE. After careful consideration, we feel that it has merit but does not fully meet PLOS ONE’s publication criteria as it currently stands. Therefore, we invite you to submit a revised version of the manuscript that addresses the points raised during the review process.

Four external reviewers have now evaluated your submission. They have identified a number of concerns that need to be carefully addressed in a revision of the manuscript. Please respond to all of the points they have raised, paying particular attention to their requests for clarifications regarding the study design and analyses and for adjustments to the phrasing and contextualisation used throughout the manuscript.

We look forward to receiving your revised manuscript.

Kind regards,

Jamie Males

Editorial Office

PLOS ONE

Journal Requirements:

4. We note you have included a table to which you do not refer in the text of your manuscript. Please ensure that you refer to Table 4 in your text; if accepted, production will need this reference to link the reader to the Table.

Reviewers' comments:

Reviewer's Responses to Questions

**Comments to the Author**

1. Is the manuscript technically sound, and do the data support the conclusions?

Reviewer #1: Partly

Reviewer #2: Partly

Reviewer #3: Partly

Reviewer #4: Yes

2. Has the statistical analysis been performed appropriately and rigorously? 

Reviewer #1: No

Reviewer #2: No

Reviewer #3: I Don't Know

Reviewer #4: Yes

3. Have the authors made all data underlying the findings in their manuscript fully available?

Reviewer #1: Yes

Reviewer #2: Yes

Reviewer #3: Yes

Reviewer #4: Yes

4. Is the manuscript presented in an intelligible fashion and written in standard English?

Reviewer #1: Yes

Reviewer #2: No

Reviewer #3: Yes

Reviewer #4: Yes

5. Review Comments to the Author

Reviewer #1: Major criticisms

1. It should be noted that data of HOMA-IR in this study may not be normally distributed (SD is higher than mean) particularly in T2D subjects (Table 1), therefore the analysis of correlations between HOMA-IR and compartmental fat or other variables by Pearson’s method may not be appropriate and may result in misleading information. Data of HOMA-IR should be log-transformed prior to analysis if Pearson’s correlation is used. Please consult statistician for this matter.

2. Given data from Table 1 and Table 2, I suspect that the majority of MetS group has T2D. (How many patients in MetS group had not been diagnosed with diabetes in this study?) If it is so, separating analysis of MetS from Non-MetS may not give you additional information since it will be similar to those of T2D group.

Minor criticisms

1. Given the SAT and VAT measurements are in part operator-dependent, the authors should specify how many radiologists involved in the study. What is the intra- or inter-observer variation of those measurements?

2. Since T2D patients were more obese than control (non-diabetes), it is interesting to see whether the differences of fat compartment between T2D and control shown in Table 4 was changed after adjustment with BMI

3. The authors may give more discussion regarding the correlations of insulin resistance and fat compartment in the quartile 4 of Table 6 whether it is specific for T2D (since T2D is the majority of population in this Q4) or it can be generalized to non-T2D as well

4. The authors should make it clear that the positive association of liver fat with insulin resistance or T2D or MetS are not independent from each other since the latter two of which have insulin resistance as a common factor.

5. In contrast with other studies, the authors should comment why, in this study, VAT was not associated with insulin resistance even in those with the most severe insulin resistance. It is interesting to note that the VAT area is quite high even in lean non-diabetic population supporting the tendency to accumulate central fat in this population.

6. Figures may not be necessary since they do not give more additional information than Tables

7. Reference format should be revised according to the Journal style.

Reviewer #2: Summary

This cross-sectional hospital-based study that included Asian-Indian adults with and without type 2 diabetes (T2D) aimed to determine the association of central abdominal fat depots (subcutaneous and visceral) and liver fat content with insulin resistance, expressed as the HOMA-IR. They reported that ectopic liver fat, but not VAT, correlated with HOMA-IR in those with T2D. In addition, they explored these associations within different levels of HOMA-IR and found that the association between ectopic liver fat and HOMA-IR was the strongest amongst the most insulin-resistant quartile. A negative association between SAT and HOMA-IR also emerged amongst those in the highest quartile. No association between VAT and HOMA-IR was found amongst individuals with and without T2D and MS. The authors concluded that these results support the adipose tissue expandability/spillover hypothesis.

This study explored an interesting and relevant question and the inclusion of a comparison group is a strength. However, the major comments are that the authors have not provided sufficient details in their methods on how the participants were selected, the precision and validity of the MRI methods and handling of the continuous variables prior to Pearson correlation analysis were not mentioned. The graphs and the tables provided overlapping results. The titles of the tables require more details in order for these tables to stand-alone. Minor comments: The abbreviations SAT and VAT refers to abdominal subcutaneous and visceral adipose tissue or fat mass. I would therefore suggest that the authors just stick to ”SAT” and “VAT” without adding fat mass after it eg. SAT fat mass or VAT fat mass. The authors will improve the reader’s experience by using consistent terms throughout the manuscript and to avoid capitalizing nouns (e.g. “Ectopic”) when it is not in the first word of a sentence.

Specific comments

Introduction

1. The second and third sentences refer to IR in the pathophysiology of T2D and MS. IR is a key feature in the pathophysiology of T2D. However, since metabolic syndrome is not a disease per se but rather a cluster of various risk factors of which IR is one (but not in always, depending on the criteria that were used), I would suggest the authors to rephrase these sentences.

2. Based on the objectives, the associations with MS is a secondary objective. I would therefore suggest the authors to first focus on T2D in the introduction and then mention MS. Also, the authors should clarify the rationale for assessing the associations of central and ectopic liver fat in those with MS. This is not clear in the introduction.

3. Sentence starting on line 10-11: Please include the comparison group. The author stated that...” they are more insulin resistant and have a higher prevalence of T2D and MS”. More than “who”? The reference for this sentence was given as Raji et al. 2001. Please cite more recent studies or even a systematic review if available.

4. Last paragraph regarding the objectives of the study: The objectives needs to be clearer. The authors need to clarify whether the first objective stated as “to find the relationship between MRI assessed abdominal fat distribution … and IR” and the second objective stated as “…to assess the relationship in different quartiles of IR in these subject” refers to those with and without type 2 diabetes.

Methodology

5. General comment – it is not clear whether medical records were used to identify the participants retrospectively or whether participants were included prospectively during visits at the hospital and whether all the participants (T2D and non T2D, MS and non-MS) received OGTTs. Can the authors please clarify this?

Study subjects

6. Please elaborate whether the participants were recruited from one particular department eg. diabetes clinic or from different departments.

7. Line 6 in this paragraph: omit the word “normal” and just mention “individuals without T2D”

8. These individuals without T2D, were they tested for T2D and tests were negative or were they never tested. If the latter is the case, this group may contain undiagnosed T2D individuals. This should be considered in the interpretation of the findings and as a limitation

9. Details regarding how participants with MS were selected and the criteria used should be moved from the clinical and biochemical assessment section to this section. A reference should be included for the IDF definition of MS. The rationale for choosing the IDF criteria should be provided.

Clinical & Biochemical assessment

10. In line 13 in the section it is mentioned that HOMA-Beta was also calculated, however, this was not shown in any of the descriptive tables. HOMA-Beta is another key factor in the pathogenesis of T2D, together with IR. In my opinion it may even be more important than IR. Even though IR may be present, hyperglycemia will not occur as long as the beta cell function remains optimal and can compensate for the level of IR. Also, a recent review published in Diabetologia (Venkat Narayan 2020) suggested that South Asians may have reduced insulin secretory ability. It would therefore add to this paper to explore the associations of central and ectopic fat depots with B-cell function.

Radiological investigations

11. The method used for the quantification of VAT, SAT and ectopic liver fat – requires more detail. The MRI procedure that was used eg. Dixon needs to be stated. I assume the area of SAT and VAT were done on 1 slice of 3 mm thickness. This needs to be clear. How was the ROIs drawn in the liver – automatic, semi-automatic or manual? Please include the precision of the methods that were used.

Statistical analysis

12. Please include the statistical software used for the analysis.

13. The authors should provide more detail regarding the distribution of the key variables used in the correlation analysis and whether any transformation of non-normally distributed variables was done prior to the Pearson correlation analysis and ANOVA (please see comment below).

14. An ANOVA was done in Table 3 but this is not mentioned in the statistical analysis. Also, I would advise a post-hoc test to be done to determine which quartiles were significantly different from each other.

15. I would suggest that the authors use a small caps “r” to abbreviate the correlation coefficient.

Results

General characteristics

16. It is stated in the 3rd line that both groups were matched for age and sex. However, the male to female ratio is given for the whole study population. I would suggest that the authors also indicate the proportion of males and females per group (T2D vs nonT2D), either in the text or in Tables 1 and 2.

17. Since this study used the IDF definition of MS. This means that having hyperglycemia or T2D is not a compulsory criterion. Besides comparing the MS and non-MS groups it will add to this paper to also mention the proportion of people classified as MS that have diabetes and hyperglycemia (fasting glucose >100mg/dL).

18. A visual presentation of the association between HOMA-IR with VAT, SAT and liver fat as continuous variables, for the whole group, by T2D status and by MS status, would be better. Chart 1 is just a repeat of Table 5. I would suggest showing the graphs with continuous variables and the correlation coefficient and p-value indicated on the graph. This will replace the current Table 5 and Chart 1.

19. Graph 1: “Correlation of HOMA-IR to fat depot on each HOMA IR Quartiles and Chart 3 as well as Table 6 provide the same information. I would suggest keeping the table and to omit the 2 graphs.

20. Table 3 and 6: Do these tables include all subjects (n=160) or only those with diabetes (n=80)? I suggest that the authors clarify this in the title of the table and include the number of participants overall and in each quartile.

Discussion

21. The authors should consider the following as potential reasons for finding no association between VAT and HOMA IR in this study:

- Association may not be linear – a curvilinear association has been described such that at higher VAT levels, no changes in IR occurs – which may explain the lack of association amongst those with higher VAT.

- HOMA-IR reflects liver insulin sensitivity and therefore the close association with hepatic fat is understandable but VAT may be more important for peripheral insulin sensitivity? Can this be a shortcoming of using HOMA-IR instead of measures of whole-body or peripheral insulin sensitivity? This should be discussed.

- Precision of measuring VAT vs ectopic liver fat could this have influenced the results? It is therefore important to state the precision and validity of the methods used to measure the fat fraction in these depots in the methodology section and it can be reiterated in the discussion.

Conclusion

Line 3 in this section: Be careful not to use the word “causative” to explain the relationship between liver ectopic fat and IR. This is a cross-sectional study and causation cannot be implied.

Reviewer #3: The paper concerns the interesting topic of assessing the correlation between ectopic liver fat, subcutaneous and abdominal visceral fat and insulin resistance, type 2 diabetes and metabolic syndrome.

Although the subject of the work itself is interesting, however, it requires additional information and numerous corrections.

1. 1. In the 'methodology' part, you can find information: ‘Ectopic liver fat was measured using liver intensity on Osirix software’.

There is no information from which sequence/sequences the software displayed the fat content automatically - I think the methodology should be described in more detail

2. 160 people were included in the study, 80 with type 2 diabetes and 80 without diabetes (Table 1). In Table 2, the 'study population' is only 158 people - what happened to the other two people?

3. Insulin resistance, i.e. the lack of tissue sensitivity to insulin, is a common metabolic disorder that can lead to the development of type 2 diabetes. As I understand it, the study assessed 80 people with already developed type 2 diabetes, not only insulin resistance, and 80 healthy people. Thus, the paper did not concern the relationship between insulin resistance and the amount of subcutaneous, visceral and ectopic liver fat but the relationship between type 2 diabetes and the quantity of subcutaneous, visceral and ectopic hepatic fat. Please clarify.

4. What were the HOMA-IR values for people with insulin resistance - as long as the study included people with insulin resistance solely and not only those with developed diabetes?

If the study included only people with already developed type 2 diabetes and healthy control subjects, the entire structure of the paper should be rebuilt.

5. There is no information on how many and what components of the metabolic syndrome (except diabetes) were found in the group of patients with MS (Table 2).

6. Table 2: Patients with MS (metabolic syndrome) accounted for 67 of 158 and without MS 91 of 158. So there are also patients with T2D in the 'without MS' column. This is misleading, I think this table should be reworked.

7. There are no citations and descriptions for charts 1, 2 and 3 in the text.

8. Table 4 is not cited in the text of the paper.

9. A broader discussion of tables and graphs in the text would improve the readability, comprehensibility, and, thus, the value of the entire work.

10. The citation brackets should be standardized - authors use both [] and ().

11. There is no abbreviation list.

12. References: use 6-8, 10-13, 16-20, not '(6,7,8,10,11,12,13,16,17,18,19,20)'.

Reviewer #4: The authors have studied the association between insulin resistance vs. subcutaneous, visceral, and liver fat in different quartiles of the HOMA-IR in 160 Asian-Indian individuals. According to their results, in those with Type 2 diabetes and metabolic syndrome, a positive correlation between liver fat and IR was observed. This association was stronger in higher quantiles of the HOMA-IR. Interestingly, they observed a negative association between IR and SAT. They have conducted proper statistical analyses and have discussed their results nicely in the discussion.

Minors:

-The small sample size is a limitation, especially regarding quartile analyses. Therefore it is good to mention it as a caveat and the results to be reported with more caution.

-In the abstract result section, stating the reciprocal relation between IR and SAT does not seem correct, as this will require bidirectional causal analysis. Perhaps the authors meant an inverse relation!

-Due to multiple hypotheses testing, control for false discovery rate needs to be done (e.g., reporting FDR corrected p-values).

-In the introduction, considering MRI and CT as the gold standard is incorrect, and they have their own limitations (e.g., cost and low precision). Even biopsy, which is usually considered the gold standard, has its caveat of being invasive.

-As the study was conducted during COVID, good to mention whether it did affect the implementation of their research or not. Also, as one of their exclusion criteria was the presence of an infection, were COVID-associated infections recorded? It will be a great addition to their study if they have such registered data. For instance, whether their measurements were recorded before having COVID or after.

-Spell out the word in the first occurrence (e.g., OGTT).

-They have several blood biomarkers measured (e.g., HDL, LDL, VLDL, and TG); why not include their raw value in the analyses?

- In tables, it is good to highlight the significant ones.

-Distribution plots are a good addition, and they will help with identifying whether normalization is required or not. Scatter plots by fitting a linear model into it is pretty informative.

-A Heatmap plot among all the variables also will make it easier to follow the results.

6. PLOS authors have the option to publish the peer review history of their article (what does this mean?). If published, this will include your full peer review and any attached files.

Reviewer #1: No

Reviewer #2: No

Reviewer #3: No

Reviewer #4: **Yes: **Naeimeh Atabaki Pasdar

---

## [Author Response · Author response to Decision Letter 0]

8 Nov 2022

Comments assessed and incorporated in revised script, Response if needed will be given if question/comment arises on appraisal of revised script

---

## [Decision Letter · Decision Letter 1]

7 Feb 2023

PONE-D-22-13467R1It is the ectopic liver fat, not the abdominal visceral fat that shows association with insulin resistance, Type-2 Diabetes and metabolic syndrome in Asian Indians.PLOS ONE

Dear Dr. Mathur,

Thank you for submitting your manuscript to PLOS ONE. After careful consideration, we feel that it has merit but does not fully meet PLOS ONE’s publication criteria as it currently stands. Therefore, we invite you to submit a revised version of the manuscript that addresses the points raised during the review process.

Your manuscript has been evaluated by two of the previous reviewers, and their comments are available below. The reviewers have raised a number of concerns that need attention. In particular, both reviewers find that your manuscript has not been fully revised to address each of the reviewers' comments on the previous version, and they have requested further clarification of methodological details and discussion of the limitations of your study. Please ensure that each of your responses to the reviewers' comments for this version and the previous version of your manuscript are accompanied by corresponding changes to your manuscript text. Please also ensure you have addressed each of the reviewers' comments in full when revising your manuscript. We also ask you to revise your Conclusions section to not use the word 'causative' to explain the relationship between liver ectopic fat and IR (as requested by Reviewer #2 of the previous version). As this is a cross-sectional study, causation cannot be implied.

We look forward to receiving your revised manuscript.

Kind regards,

Hugh Cowley

Staff Editor

PLOS ONE

Reviewers' comments:

Reviewer's Responses to Questions

**Comments to the Author**

1. If the authors have adequately addressed your comments raised in a previous round of review and you feel that this manuscript is now acceptable for publication, you may indicate that here to bypass the “Comments to the Author” section, enter your conflict of interest statement in the “Confidential to Editor” section, and submit your "Accept" recommendation.

Reviewer #1: (No Response)

Reviewer #4: (No Response)

2. Is the manuscript technically sound, and do the data support the conclusions?

Reviewer #1: Partly

Reviewer #4: Partly

3. Has the statistical analysis been performed appropriately and rigorously? 

Reviewer #1: Yes

Reviewer #4: (No Response)

4. Have the authors made all data underlying the findings in their manuscript fully available?

Reviewer #1: Yes

Reviewer #4: (No Response)

5. Is the manuscript presented in an intelligible fashion and written in standard English?

Reviewer #1: Yes

Reviewer #4: (No Response)

6. Review Comments to the Author

Reviewer #1: 1. Abstract: the Results section was not yet revised.

1.1 The sentence “In T2D and MS, HOMA-IR showed a moderate strong correlation with ectopic liver fat…but not SAT and VAT” is not true. In Table 5, the log-transformed HOMA-IR had no correlation with liver fat in both T2D and MS.

1.2 The next following sentence “On analyzing the correlation of abdominal adipose compartment fat mass with IR…., ectopic liver fat shows a positive correlation with HOMA-IR (r=0.52, p<0.05)” is not correct since the correlation of log-transformed HOMA-IR and liver fat was 0.363 and was not significant (Table 5).

1.3 Likewise the sentence “SAT shows an inverse relation with IR with a statistically significant negative correlation in the highest quartile” is also not true given the p values of 0.05 in log-transformed HOMA-IR and SAT analysis (Table 6)

1.4 The Conclusion should be revised in accordance with the new results

2. Methodology

2.1 The method of statistical work which include the log-transformed of HOMA-IR data prior to Pearson’s correlation analysis and ANOVA (Table 3) should be mentioned in methodology section

2.2 The precision of manually-measured SAT, VAT area by the radiologist for example intra-observer CV of measurements should be reported

3. Results

The correlation results (r) and their significance level (p) between HOMA-IR and fat compartments were changed after log-transformed data of HOMA-IR was applied. Therefore the r and p values should be revised accordingly

4. Discussion

4.1 The statements “…in its highest quartile, HOMA-IR showed a significantly negative correlation with SAT” and the following “SAT fat mass as a protective role in the pathogenesis of IR” of which it appeared in several occasions in the Discussion and Conclusion was not quite right since the p value was 0.05. It is safer if the authors rephrase the statements “…HOMA-IR tended to show a negative correlation with SAT” and “SAT may have a protective role…”

4.2 In the second paragraph of Discussion, the authors quoted the significant correlation (r) results of HOMA-IR and liver fat in the whole (r=0.52) T2D (r=0.609) and MS (r=0.673) population, of which they are not correct values. In log-transformed HOMA-IR data, those were 0.363, 0.46 and 0.493 respectively (Table 5). All are not significant.

4.3 The authors should add a little more discussion why VAT was not correlated with HOMA-IR in this study of which it is in contrast with several others. It should be noted that log-transformed HOMA-IR and VAT tended to be correlated at the highest quartile of IR (p=0.078)

5. Tables and Figures

5.1 In Table 5 and 6 regarding the correlation of HOMA-IR and fat depots, only the log-transformed HOMA-IR data should be reported.

5.2 Figure1 is not necessary, it repeated what appeared in Table 6.

Reviewer #4: The authors have not put enough effort into addressing the comments. Small sample size, multiple hypothesis testing, and not correcting for that are important caveats that need to be at least discussed and acknowledged. Visualizing the distributions of the data or a correlation matrix/Heatmap plot among the variables would greatly help understand the data and interpret the results.

7. PLOS authors have the option to publish the peer review history of their article (what does this mean?). If published, this will include your full peer review and any attached files.

Reviewer #1: No

Reviewer #4: **Yes: **Naeimeh Atabaki-Pasdar

---

## [Author Response · Author response to Decision Letter 1]

4 Apr 2023

Response to Reviewers’ comment - Manuscript no. PONE-D-22-13467

Title---It is the ectopic liver fat, not the abdominal visceral fat that shows association with insulin resistance, Type-2 Diabetes and metabolic syndrome in Asian Indians. 

Reviewer#1: Major criticisms

1. It should be noted that data of HOMA-IR in this study may not be normally distributed (SD is higher than mean) particularly in T2D subjects (Table 1), therefore the analysis of correlations between HOMA-IR and compartmental fat or other variables by Pearson’s method may not be appropriate and may result in misleading information. Data of HOMA-IR should be log-transformed prior to analysis if Pearson’s correlation is used. Please consult statistician for this matter.

Response: 

As per comments, log transformation of HOMA-IR has been done and included in analysis in revised manuscript. Although, results are nearly similar.

2. Given data from Table 1 and Table 2, I suspect that the majority of MetS group has T2D. (How many patients in MetS group had not been diagnosed with diabetes in this study?) If it is so, separating analysis of MetS from Non-MetS may not give you additional information since it will be similar to those of T2D group.

Response: 

Out of 67 people with metabolic syndrome, 58 had DM, while only 21 persons with diabetes were in non-metabolic syndrome group (21/97). We agree with your interpretation.

Minor criticisms

1. Given the SAT and VAT measurements are in part operator-dependent, the authors should specify how many radiologists involved in the study. What is the intra- or inter-observer variation of those measurements?

Response: 

Only one investigator was involved in SAT and VAT measurements and he was blinded from patient clinical and biochemical data.

2. Since T2D patients were more obese than control (non-diabetes), it is interesting to see whether the differences of fat-compartment between T2D and control shown in Table 4 was changed after adjustment with BMI.

Response: 

That table has now Table 3 in current version. As suggested, we have also included BMI-adjusted P values also for all the groups across all the three adipose sites. We could see that no major change in the overall results, when study data were adjusted to BMI.

3. The authors may give more discussion regarding the correlations of insulin resistance and fat compartment in the quartile 4 of Table 6 whether it is specific for T2D (since T2D is the majority of population in this Q4) or it can be generalized to non-T2D as well.

Response: 

Thanks for advising on this important point. It has been discussed in detail in the discussion section. As it has been discussed that fat re-distribution from SAT to ectopic liver compartment is true for T2D and MetS group, majority of them falls in Q4. This table has now become Table 5 in current version. 

4. The authors should make it clear that the positive association of liver fat with insulin resistance or T2D or MetS are not independent from each other since the latter two of which have insulin resistance as a common factor.

 Response: 

Thanks for pointing it out. The same has been mentioned in discussion (lines 196-202)

5. In contrast with other studies, the authors should comment why, in this study, VAT was not associated with insulin resistance even in those with the most severe insulin resistance. It is interesting to note that the VAT area is quite high even in lean non-diabetic population supporting the tendency to accumulate central fat in this population.

Response: 

On re-analysis of the data using log HOMA-IR (instead of HOMA-IR), it showed borderline (though statistically insignificant) positive association with the VAT mass, particularly in its highest quartile. Therefore, the whole manuscript has been revised accordingly.

6. Figures may not be necessary since they do not give more additional information than tables.

Response: 

Figures are now removed as per comment.

7. Reference format should be revised according to the Journal style.

Response: 

References format has been revised.

Reviewer #2:

Introduction

1. The second and third sentences refer to IR in the pathophysiology of T2D and MS. IR is a key feature in the pathophysiology of T2D. However, since metabolic syndrome is not a disease per se but rather a cluster of various risk factors of which IR is one (but not in always, depending on the criteria that were used), I would suggest the authors to rephrase these sentences.

Response: 

Sentence has been rephrased (line 41).

2.Based on the objectives, the associations with MS is a secondary objective. I would therefore suggest the authors to first focus on T2D in the introduction and then mention MS. Also, the authors should clarify the rationale for assessing the associations of central and ectopic liver fat in those with MS. This is not clear in the introduction.

Response: 

We agreed that discussion and data focused to T2D. Central (non-ectopic fat i.e., VAT, and liver fat) were different in occurrence and distribution and different stage of patho-biogenesis; hence discussed.

3.Sentence starting on line 10-11: Please include the comparison group. The author stated that...” they are more insulin resistant and have a higher prevalence of T2D and MS”. More than “who”? The reference for this sentence was given as Raji et al. 2001. Please cite more recent studies or even a systematic review if available.????

Response: 

Systematic reviews are not available to best of authors knowledge. Please let us know if any.

4.Last paragraph regarding the objectives of the study: The objectives need to be clearer. The authors need to clarify whether the first objective stated as “to find the relationship between MRI assessed abdominal fat distribution … and IR” and the second objective stated as “…to assess the relationship in different quartiles of IR in these subject” refers to those with and without type 2 diabetes.

Response: 

Corrected as per your comment.

Methodology

5. General comment – it is not clear whether medical records were used to identify the participants retrospectively or whether participants were included prospectively during visits at the hospital and whether all the participants (T2D and non T2D, MS and non-MS) received OGTTs. Can the authors please clarify this? 

Response: 

Study subjects’ medical record along with history, were used to identify participants and those not having T2D confirmed with OGTT as mentioned.

6. Please elaborate whether the participants were recruited from one particular department eg. diabetes clinic or from different departments.

Response: 

Participants were recruited from Single department.

7. Line 6 in this paragraph: omit the word “normal” and just mention “individuals without T2D”.

Response: 

Corrected as per suggestion.

8. These individuals without T2D, were they tested for T2D and tests were negative or were they never tested. If the latter is the case, this group may contain undiagnosed T2D individuals. This should be considered in the interpretation of the findings and as a limitation.

Response: 

Those without T2D were tested with OGTT to confirm status as mentioned in methodology.

9. Details regarding how participants with MS were selected and the criteria used should be moved from the clinical and biochemical assessment section to this section. A reference should be included for the IDF definition of MS. The rationale for choosing the IDF criteria should be provided. Clinical & Biochemical assessment. 

Response: 

IDF harmonizing criteria was used as it has specific consideration to our population (South East Asia). Reference mentioned.

10. In line 13 in the section it is mentioned that HOMA-Beta was also calculated, however, this was not shown in any of the descriptive tables. HOMA-Beta is another key factor in the pathogenesis of T2D, together with IR. In my opinion it may even be more important than IR. Even though IR may be present, hyperglycemia will not occur as long as the beta cell function remains optimal and can compensate for the level of IR. Also, a recent review published in Diabetologia (Venkat Narayan 2020) suggested that South Asians may have reduced insulin secretory ability. It would therefore add to this paper to explore the associations of central and ectopic fat depots with B-cell function.

Response: 

South Asian were traditionally considered to have reduced secretory capacity but recent clusters studies have shown different phenotypes ranging from insulin resistance to insulin deficient and combined insulin resistance and deficiency. As the aim of study was to explore insulin resistance rather than Diabetes pathogenesis; HOMA-beta analysis not taken into analysis in this study.

Radiological investigations

11. The method used for the quantification of VAT, SAT and ectopic liver fat – requires more detail. The MRI procedure that was used eg. Dixon needs to be stated. I assume the area of SAT and VAT were done on 1 slice of 3 mm thickness. This needs to be clear. How was the ROIs drawn in the liver – automatic, semi-automatic or manual? Please include the precision of the methods that were used.

Response: 

Dixon method used by single radiologist with ROI drawn manually.

Statistical analysis

12. Please include the statistical software used for the analysis.

Response: 

SPSS26 was software used for the analysis.

13. The authors should provide more detail regarding the distribution of the key variables used in the correlation analysis and whether any transformation of non-normally distributed variables was done prior to the Pearson correlation analysis and ANOVA (please see comment below).

Response: 

Done

14. An ANOVA was done in Table 3 but this is not mentioned in the statistical analysis. Also, I would advise a post-hoc test to be done to determine which quartiles were significantly different from each other.

Response: 

Done

15. I would suggest that the authors use a small caps “r” to abbreviate the correlation coefficient.

Response: 

Done

Results

General characteristic

16. It is stated in the 3rd line that both groups were matched for age and sex. However, the male to female ratio is given for the whole study population. I would suggest that the authors also indicate the proportion of males and females per group (T2D vs nonT2D), either in the text or in Tables 1 and 2.

Response: 

Done

17. Since this study used the IDF definition of MS. This means that having hyperglycemia or T2D is not a compulsory criterion. Besides comparing the MS and non-MS groups it will add to this paper to also mention the proportion of people classified as MS that have diabetes and hyperglycemia (fasting glucose > 100mg/dL).

Response: 

58 out of 67 people with MS were having T2D and out of 91 people without MS 21 were suffering from T2D.

18. A visual presentation of the association between HOMA-IR with VAT, SAT and liver fat as continuous variables, for the whole group, by T2D status and by MS status, would be better. Chart 1 is just a repeat of Table 5. I would suggest showing the graphs with continuous variables and the correlation coefficient and p-value indicated on the graph. This will replace the current Table 5 and Chart 1.

Response: 

Done

19. Graph 1: “Correlation of HOMA-IR to fat depot on each HOMA IR Quartiles and Chart 3 as well as Table 6 provide the same information. I would suggest keeping the table and to omit the 2 graphs.

Response: 

Done

20. Table 3 and 6: Do these tables include all subjects (n=160) or only those with diabetes (n=80)? I suggest that the authors clarify this in the title of the table and include the number of participants overall and in each quartile.

Response: 

Whole population is included.

Discussion

21. The authors should consider the following as potential reasons for finding no association between VAT and HOMA IR in this study:

- Association may not be linear – a curvilinear association has been described such that at higher VAT levels, no changes in IR occurs – which may explain the lack of association amongst those with higher VAT.

- HOMA-IR reflects liver insulin sensitivity and therefore the close association with hepatic fat is understandable but VAT may be more important for peripheral insulin sensitivity? Can this be a shortcoming of using HOMA-IR instead of measures of whole-body or peripheral insulin sensitivity? This should be discussed.

- Precision of measuring VAT vs ectopic liver fat could this have influenced the results? It is therefore important to state the precision and validity of the methods used to measure the fat fraction in these depots in the methodology section and it can be reiterated in the discussion.

Response: 

It is done

Reviewer #3

1. In the 'methodology' part, you can find information: ‘Ectopic liver fat was measured using liver intensity on Osirix software’.

There is no information from which sequence/sequences the software displayed the fat content automatically - I think the methodology should be described in more detail.

Response: 

Dixon method used on Osirix software by single investigator, and as per your suggestion included in methodology.

2. 160 people were included in the study, 80 with type 2 diabetes and 80 without diabetes (Table 1). In Table 2, the 'study population' is only 158 people - what happened to the other two people?

Response: 

While study population was 160, 2 subjects have data missing on lipid profile hence not included in metabolic syndrome analysis. So, when talking about metabolic syndrome wise population in analysis reduced to 158.

3. Insulin resistance, i.e. the lack of tissue sensitivity to insulin, is a common metabolic disorder that can lead to the development of type 2 diabetes. As I understand it, the study assessed 80 people with already developed type 2 diabetes, not only insulin resistance, and 80 healthy people. Thus, the paper did not concern the relationship between insulin resistance and the amount of subcutaneous, visceral and ectopic liver fat but the relationship between type 2 diabetes and the quantity of subcutaneous, visceral and ectopic hepatic fat. Please clarify.

Response: 

As we also analysed relationship in people without diabetes (confirmed by OGTT), along with diabetics so we concern with IR rather than T2D.

4. What were the HOMA-IR values for people with insulin resistance - as long as the study included people with insulin resistance solely and not only those with developed diabetes?

Response: 

Rebuilt and done

5. There is no information on how many and what components of the metabolic syndrome (except diabetes) were found in the group of patients with MS (Table 2).

Response: 

As we did not stratify according to number of features of MS, we took 3 or more features as per IDF harmonizing criteria for MS.

6. Table 2: Patients with MS (metabolic syndrome) accounted for 67 of 158 and without MS 91 of 158. So there are also patients with T2D in the 'without MS' column. This is misleading, I think this table should be reworked.

Response: 

Yes, 58 out of 67 people with MS were diabetic while 21 out of 97 non-MS people were diabetic. As IDF criteria diabetes is not compulsory hence it is not misleading.

7. There are no citations and descriptions for charts 1, 2 and 3 in the text.

Response: 

Done

8. Table 4 is not cited in the text of the paper.

Response: 

Done

9. A broader discussion of tables and graphs in the text would improve the readability, comprehensibility, and, thus, the value of the entire work.

Response: 

Done

10. The citation brackets should be standardized - authors use both [] and ()

Response: 

Done

11. There is no abbreviation list.

Response: 

Done

12. References: use 6-8, 10-13, 16-20, not '(6,7,8,10,11,12,13,16,17,18,19,20)'

Response: 

Done

Reviewer #4:

Minors:

-The small sample size is a limitation, especially regarding quartile analyses. Therefore, it is good to mention it as a caveat and the results to be reported with more caution.

Response: 

These limitations of the study are mentioned in the discussion section of the manuscript.

-In the abstract result section, stating the reciprocal relation between IR and SAT does not seem correct, as this will require bidirectional causal analysis. Perhaps the authors meant an inverse relation!

Response: 

Thanks for pointing it out. Appropriate correction has been made.

-Due to multiple hypotheses testing, control for false discovery rate needs to be done (e.g., reporting FDR corrected p-values).

Response: 

We calculated correlations over aggregate sample values not for each individual separately so FDR correction would likely not required. 

 -In the introduction, considering MRI and CT as the gold standard is incorrect, and they have their own limitations (e.g., cost and low precision). Even biopsy, which is usually considered the gold standard, has its caveat of being invasive.

Response: 

Thanks for pointing it out. Appropriate correction has been made.

-As the study was conducted during COVID, good to mention whether it did affect the implementation of their research or not. Also, as one of their exclusion criteria was the presence of an infection, were COVID-associated infections recorded? It will be a great addition to their study if they have such registered data. For instance, whether their measurements were recorded before having COVID or after.

Covid related information was not noted.

-Spell out the word in the first occurrence (e.g., OGTT).

-They have several blood biomarkers measured (e.g., HDL, LDL, VLDL, and TG); why not include their raw value in the analyses?

- In tables, it is good to highlight the significant ones.

Thanks for this suggestion. It is done.

-Distribution plots are a good addition, and they will help with identifying whether normalization is required or not. Scatter plots by fitting a linear model into it is pretty informative.

-A Heatmap plot among all the variables also will make it easier to follow the results.

Response: 

Heatmap has now been added. 

Second review

Reviewer #1: 

1. Abstract: The Results section was not yet revised.

1.1 The sentence “In T2D and MS, HOMA-IR showed a moderate strong correlation with ectopic liver fat…but not SAT and VAT” is not true. In Table 5, the log-transformed HOMA-IR had no correlation with liver fat in both T2D and MS.

Response: 

Thanks for pointing out this important point. The whole data is re-analysed with log HOMA-IR instead of HOMA-IR and corrections are made accordingly throughout the manuscript right from title to inference. The ectopic liver fat still shows significant correlation with Log HOMA-IR. VAT show borderline correlation, which is though statistically not significant, but is marginal only.

1.2 The next following sentence “On analysing the correlation of abdominal adipose compartment fat mass with IR…., ectopic liver fat shows a positive correlation with HOMA-IR (r=0.52, p < 0.05)” is not correct since the correlation of log-transformed HOMA-IR and liver fat was 0.363 and was not significant (Table 5).

Response: 

Among diabetics and MetS group the correlation coefficients are 0.46 and 0.493 respectively and they are statistically significant. As participants in this study were recruited in T2D and non T2D groups separately, therefore analysis of combined group has been removed from analysis. 

1.3 Likewise the sentence “SAT shows an inverse relation with IR with a statistically significant negative correlation in the highest quartile” is also not true given the p values of 0.05 in log-transformed HOMA-IR and SAT analysis (Table 6)??

Response: 

It is P < 0.05 and has been corrected. 

1.4 The Conclusion should be revised in accordance with the new results

Response: 

The conclusions have been revised accordingly.

2. Methodology

2.1 The method of statistical work which include the log-transformed of HOMA-IR data prior to Pearson’s correlation analysis and ANOVA (Table 3) should be mentioned in methodology section??

2.2 The precision of manually-measured SAT, VAT area by the radiologist for example intra-observer CV of measurements should be reported

Response: 

Fat analysis was done by a single person.

3. Results

The correlation results (r) and their significance level (p) between HOMA-IR and fat compartments were changed after log-transformed data of HOMA-IR was applied. Therefore, the r and p values should be revised accordingly

Response: 

The values have been revised accordingly

4. Discussion

4.1 The statements “…in its highest quartile, HOMA-IR showed a significantly negative correlation with SAT” and the following “SAT fat mass as a protective role in the pathogenesis of IR” of which it appeared in several occasions in the Discussion and Conclusion was not quite right since the p value was 0.05. It is safer if the authors rephrase the statements “…HOMA-IR tended to show a negative correlation with SAT” and “SAT may have a protective role…”

4.2 In the second paragraph of Discussion, the authors quoted the significant correlation (r) results of HOMA-IR and liver fat in the whole (r=0.52) T2D (r=0.609) and MS (r=0.673) population, of which they are not correct values. In log-transformed HOMA-IR data, those were 0.363, 0.46 and 0.493 respectively (Table 5). All are not significant.

Response: 

The log transformed HOMA-IR are used in correlation analysis and accordingly title, results and discussion and conclusions are modified.

4.3 The authors should add a little more discussion why VAT was not correlated with HOMA-IR in this study of which it is in contrast with several others. It should be noted that log-transformed HOMA-IR and VAT tended to be correlated at the highest quartile of IR (p=0.078)

This is a very important and the whole manuscript has now been revised accordingly. Thanks for pointing it out. This inconsistent finding is discussed in the discussion section lines 248-253.

5. Tables and Figures

5.1 In Table 5 and 6 regarding the correlation of HOMA-IR and fat depots, only the log-transformed HOMA-IR data should be reported.

Response: 

Only log transformed HOMA-IR has been reported in the table.

5.2 Figure1 is not necessary, it repeated what appeared in Table 6.

Response: 

The figure has been removed.

Reviewer #4: 

The authors have not put enough effort into addressing the comments. Small sample size, multiple hypothesis testing, and not correcting for that are important caveats that need to be at least discussed and acknowledged. Visualizing the distributions of the data or a correlation matrix/Heatmap plot among the variables would greatly help understand the data and interpret the results.??

Response: 

Thanks for the comments. The manuscript has now been thoroughly revised and previous suggestions of reviewers that were not addressed in the second revision are now incorporated in this third revision. The correlation matrix is added as figure 1 in the results. The limitations mentioned above are incorporated in the discussion (lines 256-60)

We have made substantial improvements as per comments of our revered reviewers and hope the current version of our manuscript would come up to the expectation of them. We also enclosed a correlation plot to feature relationships across variable of interest.

---

## [Decision Letter · Decision Letter 2]

4 Jun 2023

PONE-D-22-13467R2It is the ectopic liver fat, not the abdominal visceral fat that shows association with insulin resistance, Type-2 Diabetes and metabolic syndrome in Asian Indians.PLOS ONE

Dear Dr. Mathur,

Thank you for submitting your manuscript to PLOS ONE. After careful consideration, we feel that it has merit but does not fully meet PLOS ONE’s publication criteria as it currently stands. Therefore, we invite you to submit a revised version of the manuscript that addresses the points raised during the review process.Please review the title, methodology and conclusion as per comments belowOther specific feedback is provided below.Please submit your revised manuscript by Jul 19 2023 11:59PM. If you will need more time than this to complete your revisions, please reply to this message or contact the journal office at plosone@plos.org. Please include the following items when submitting your revised manuscript:A rebuttal letter that responds to each point raised by the academic editor and reviewer(s). You should upload this letter as a separate file labeled 'Response to Reviewers'.A marked-up copy of your manuscript that highlights changes made to the original version. You should upload this as a separate file labeled 'Revised Manuscript with Track Changes'.An unmarked version of your revised paper without tracked changes. You should upload this as a separate file labeled 'Manuscript'.

We look forward to receiving your revised manuscript.

Kind regards,

Fredirick Lazaro mashili, MD, PhD

Academic Editor

PLOS ONE

Additional Editor Comments:

Put together, the title, methodology and conclusion do not clearly tie.

Visceral adipose tissue (VAT) refers to the fat that surrounds the body's internal organs, particularly those in the abdominal region such as the liver, pancreas, and intestines also including omental fat. It's one of the two main types of abdominal fat, with the other being subcutaneous adipose tissue which lies just underneath the skin. This means that ectopic liver fat could also be part of VAT. As per methodological description, what was described as VAT was literally omental fat, none of the other depots that makes up VAT were measured (or were measured but not presented and described). Based on that the title and conclusion of this study do not reflect what was exactly done. The authors needed to include the other depots making up VAT for them to conclude that it is only the ectopic liver fat and not other depots that are associated with insulin resistance. Additionally, the authors should directly say omental fat rather than VAT, if what was measured and referred to as VAT was only omental fat. Furthermore, what is described as subcutaneous fat was only measured around the abdomen. Using SAT is very confusing given that only abdominal SAT was measured. The authors need to be very specific.

It is also important for the authors to be very clear in the use of terms such as ectopic fat. Ectopic fat may refer to the deposition of triglycerides within cells of non-adipose tissue that normally contain only small amounts of fat. These tissues include the liver (resulting in fatty liver disease), muscles, heart, and pancreas. Ectopic fat accumulation is believed to occur when the storage capacity of healthy adipose tissue (like subcutaneous fat) is exceeded, leading to an overflow of excess free fatty acids into the circulation and uptake into these other tissues.

Therefore, while visceral fat is a type of adipose tissue located in a specific area (around the internal organs in the abdominal cavity), ectopic fat refers more to a pathophysiological condition where fat is stored in cells of tissues that usually do not store large amounts of fat. Based on this fact, it might be that what is associated with insulin resistance is FATTY LIVER and not what they described as ectopic liver fat. Reflecting on these confusions the authors need to clearly elaborate and justify the use of terms they decide to use with much clarity.

Reviewers' comments:

Reviewer's Responses to Questions

**Comments to the Author**

Reviewer #1: (No Response)

2. Is the manuscript technically sound, and do the data support the conclusions?

Reviewer #1: Partly

3. Has the statistical analysis been performed appropriately and rigorously? 

Reviewer #1: Yes

4. Have the authors made all data underlying the findings in their manuscript fully available?

Reviewer #1: (No Response)

5. Is the manuscript presented in an intelligible fashion and written in standard English?

Reviewer #1: Yes

6. Review Comments to the Author

Reviewer #1: 1. Abstract

Results: line 33-35; please correct the r and p values of the correlation between SAT and IR in highest quartile. It should be r = -0.2843 and p = 0.05

2. Methodology

Analysis of variance for between groups comparison should be included in the statistical analysis

3. Results

3.1 line 174-176: the r-value of the correlation between HOMA-IR and ectopic liver fat in subjects with MetS was 0.673, of which it was in disagreement with that of 0.493 as shown in Table 3. Which number is the correct one? It should be noted that the r-value of 0.493 was not significant by the previous version of the manuscript (Table 5). The p-value of the correlation was also in disagreement with that shown in Table 3.

3.2 Likewise, the r-value of the correlation between HOMA-IR and VAT was incorrect.

3.3 Please correct the symbol used to indicate the p-values in Table 3, for examples p = 0.97 not p <0.97

3.4 Line 181: the comparisons of subject characteristics according to IR quintiles was in fact, shown in Table 5 not Table 4. Likewise, in line 188, the correlation of fat compartments with IR quintiles was shown in Table 4, not Table 5

3.5 Line 191-193: the statement “SAT showed statistically significant negative correlation with log HOMA-IR in the highest quintile of IR” is over-claimed since the p-value was not significant (p=0.05) as shown in Table 4

4. Discussions and Conclusions

It appears that the authors over-claimed SAT as a protective factor against IR in several occasions, for examples in line 207-209, 247-248, 266-268 although it was not supported by the study results.

7. PLOS authors have the option to publish the peer review history of their article (what does this mean?). If published, this will include your full peer review and any attached files.

Reviewer #1: No

---

## [Author Response · Author response to Decision Letter 2]

24 Jun 2023

Response to Reviewer’s comment - Manuscript no. PONE-D-22-13467

It is the Fatty Liver, Not the Adipose Depots Surrounding the Intraperitoneal Organs That Shows Association with Insulin Resistance in Asian Indian having T2DM & Metabolic Syndrome.

Editor’s comment:

Put together, the title, methodology and conclusion do not clearly tie.

Visceral adipose tissue (VAT) refers to the fat that surrounds the body's internal organs, particularly those in the abdominal region such as the liver, pancreas, and intestines also including omental fat. It's one of the two main types of abdominal fat, with the other being subcutaneous adipose tissue which lies just underneath the skin. This means that ectopic liver fat could also be part of VAT. As per methodological description, what was described as VAT was literally omental fat, none of the other depots that makes up VAT were measured (or were measured but not presented and described). Based on that the title and conclusion of this study do not reflect what was exactly done. The authors needed to include the other depots making up VAT for them to conclude that it is only the ectopic liver fat and not other depots that are associated with insulin resistance. Additionally, the authors should directly say omental fat rather than VAT, if what was measured and referred to as VAT was only omental fat. Furthermore, what is described as subcutaneous fat was only measured around the abdomen. Using SAT is very confusing given that only abdominal SAT was measured. The authors need to be very specific.

It is also important for the authors to be very clear in the use of terms such as ectopic fat. Ectopic fat may refer to the deposition of triglycerides within cells of non-adipose tissue that normally contain only small amounts of fat. These tissues include the liver (resulting in fatty liver disease), muscles, heart, and pancreas. Ectopic fat accumulation is believed to occur when the storage capacity of healthy adipose tissue (like subcutaneous fat) is exceeded, leading to an overflow of excess free fatty acids into the circulation and uptake into these other tissues.

Therefore, while visceral fat is a type of adipose tissue located in a specific area (around the internal organs in the abdominal cavity), ectopic fat refers more to a pathophysiological condition where fat is stored in cells of tissues that usually do not store large amounts of fat. Based on this fact, it might be that what is associated with insulin resistance is FATTY LIVER and not what they described as ectopic liver fat. Reflecting on these confusions the authors need to clearly elaborate and justify the use of terms they decide to use with much clarity.

Response 

Thanks for pointing out this terminology confusion. We have now clarified this confusion and have selected new terms for the three abdominal fat / adipose compartments measured , i.e. liver fat, subcutaneous adipose tissue (SAT) and adipose tissue compartment surrounding intraperitoneal organs (S-VAT). These terms are selected in such a way that they reflect exactly what was measured in this study. It is mentioned in the methods section (line 124-138). The title, methodology and conclusions are changed accordingly.

Reviewer #1:

1. Abstract

Comment

Results: line 33-35; please correct the r and p values of the correlation between SAT and IR in highest quartile. It should be r = -0.2843 and p = 0.05

Response

It has now been corrected.

2. Methodology

Comment

Analysis of variance for between groups comparison should be included in the statistical analysis

Response

In this study, we used student T-test for parametric variables which is considered to less likely commit an error in comparison to ANOVA. Moreover we always compare two groups at a time so ANOVA was not essential. 

3. Results

Comment

3.1 line 174-176: the r-value of the correlation between HOMA-IR and ectopic liver fat in subjects with MetS was 0.673, of which it was in disagreement with that of 0.493 as shown in Table 3. Which number is the correct one? It should be noted that the r-value of 0.493 was not significant by the previous version of the manuscript (Table 5). The p-value of the correlation was also in disagreement with that shown in Table 3.

Response

Table 5 and 6 in the first version correspond to table 3 and 4 in the second version. Earlier table 5 shows correlations of HOMA-IR with different fat mass whereas in second version, table 3 shows correlation of log-HOMA-IR with BMI-adjusted fat masses. That’s why both r and p values are different (r = 0.493, P < 0.0000001) than the previous one (r = 0.673, P < 0.05). We have also updated these values between lines 181-183 in text.

Comment

3.2 Likewise, the r-value of the correlation between HOMA-IR and VAT was incorrect.

Response

Table 5 and 6 in the first version correspond to table 3 and 4 in the second version.

Comment

3.3 Please correct the symbol used to indicate the p-values in Table 3, for examples p = 0.97 not p <0.97

Response

Corrected

Comment

3.4 Line 181: the comparisons of subject characteristics according to IR quintiles was in fact, shown in Table 5 not Table 4. Likewise, in line 188, the correlation of fat compartments with IR quintiles was shown in Table 4, not Table 5

Response

Corrected

Comment

3.5 Line 191-193: the statement “SAT showed statistically significant negative correlation with log HOMA-IR in the highest quintile of IR” is over-claimed since the p-value was not significant (p=0.05) as shown in Table 4

Response

This sentence has now been updated as:

“SAT showed an almost statistically significant negative correlation with log HOMA-IR in the highest quartile of IR (r = -0.2843 P ≈ 0.05 in quartile IV)”.

Comment

4. Discussions and Conclusions

It appears that the authors over-claimed SAT as a protective factor against IR in several occasions, for examples in line 207-209, 247-248, 266-268 although it was not supported by the study results.

Response

Thanks for pointing out this important issue. The discussion and conclusions are re-written accordingly.

---

## [Decision Letter · Decision Letter 3]

24 Oct 2023

PONE-D-22-13467R3It is the Fatty Liver, Not the Adipose Depots Surrounding the Intraperitoneal Organs That Shows Association with Insulin Resistance in Asian Indian having T2DM & Metabolic Syndrome.PLOS ONE

Dear Dr. Mathur,

Thank you for submitting your manuscript to PLOS ONE. After careful consideration, we feel that it has merit but does not fully meet PLOS ONE’s publication criteria as it currently stands. Therefore, we invite you to submit a revised version of the manuscript that addresses the points raised during the review process.

Please consider revising your title to make it more general than conclusive, also use appropriate and clear terms to describe the fat depotsThis manuscript requires a major revisionPlease address all the comments from the reviewers ==============================

We look forward to receiving your revised manuscript.

Kind regards,

Fredirick Lazaro mashili, MD, PhD

Academic Editor

PLOS ONE

Additional Editor Comments (if provided):

The manuscript is still potential for publication, however it requires a major revision

1. The title is very conclusive while the supporting data is not very clear, the sample size is NOT convincing for such a conclusive title

2. Metabolic syndrome is included in the discussions and conclusion while the methodology show no inclusion of patients with metabolic syndrome

3. The description and terms used for fat depots are still confusing

Reviewers' comments:

Reviewer's Responses to Questions

**Comments to the Author**

1. If the authors have adequately addressed your comments raised in a previous round of review and you feel that this manuscript is now acceptable for publication, you may indicate that here to bypass the “Comments to the Author” section, enter your conflict of interest statement in the “Confidential to Editor” section, and submit your "Accept" recommendation.

Reviewer #5: (No Response)

Reviewer #6: (No Response)

2. Is the manuscript technically sound, and do the data support the conclusions?

Reviewer #5: Yes

Reviewer #6: Partly

3. Has the statistical analysis been performed appropriately and rigorously? 

Reviewer #5: Yes

Reviewer #6: Yes

4. Have the authors made all data underlying the findings in their manuscript fully available?

Reviewer #5: No

Reviewer #6: Yes

5. Is the manuscript presented in an intelligible fashion and written in standard English?

Reviewer #5: Yes

Reviewer #6: Yes

6. Review Comments to the Author

Reviewer #5: I have reviewed the manuscript, the following minor issues needs to be addressed.

1. P-values should be reported with uniform decimal places throughout the manuscript also there should be a consinstence in writting letter "p". To enhence readability there should be no mixing of small and capital letter

2. Authors should correct all typographical errors throughout the manuscript. forexample line 141. its written p-val

3. Authors should consider providing interpretion of all abbreviations when they are used for the first time in the text, forexample line 118. "IDF" there is subsequent definition of the abbreviation in parantheses

4. reference no 22 is incomplete.

5. I suggest authors considering to use non parametric correlation analysis, to see what direction of relationship between liver fat and IR they get. this is because there is possiblity of log transformation changing the directions of correlation relationship besides the fact that it transform non-normally distributed data into a more normal distribution.

Reviewer #6: While you have tried to address all the comments, the comments from the editor haven’t been sufficiently addressed. There is still some confusion between the conclusion made and what exactly was measured. You conclude that it is the liver fat that shows association and not visceral adipose tissue, where as the visceral adipose tissue also include liver fat. Additionally, you also include metabolic syndrome in your title, discussion and conclusion while the methodology doesn’t describe or show how patients with metabolic syndrome were selected and recruited. Furthermore, your title is too conclusive while not clearly supported by data. You would rather have written “Abdominal fat depots and their association with insulin resistance in patients with type 2 diabetes” and have the freedom to describe your findings rather than having a very conclusive title. The small sample size is also hindrance for such a conclusive title.

7. PLOS authors have the option to publish the peer review history of their article (what does this mean?). If published, this will include your full peer review and any attached files.

Reviewer #5: **Yes: **Oscar Mbembela

Reviewer #6: **Yes: **Fredirick mashili

---

## [Author Response · Author response to Decision Letter 3]

1 Nov 2023

Respected Dr Mashili,

Thank you very much for conducting the 3rd round of review for our submitted manuscript. We have gone through the comments and suggestions given by our revered Reviewers and could come up with our revised manuscript which hopefully satisfy their concerns. 

A point-by-point comments is enclosed herewith: 

Comments to the Author

4. Have the authors made all data underlying the findings in their manuscript fully available?

Reviewer #5: No

Response

All data underlying the findings in manuscript has now been fully available in Supplementary File. 

6. Review Comments to the Author

Reviewer #5: I have reviewed the manuscript, the following minor issues needs to be addressed.

1. P-values should be reported with uniform decimal places throughout the manuscript also there should be a consinstence in writting letter "p". To enhence readability there should be no mixing of small and capital letter

2. Authors should correct all typographical errors throughout the manuscript. For example, line 141. its written p-val

Response

P values have been reported with uniform two-decimal places and also following consistent writing style and Typographical errors have been rectified to best of our knowledge. 

3. Authors should consider providing interpretation of all abbreviations when they are used for the first time in the text, for example line 118. "IDF" there is subsequent definition of the abbreviation in parentheses

Response

Resolved

4. reference no 22 is incomplete.

Response

In is now complete 

5. I suggest authors considering to use non-parametric correlation analysis, to see what direction of relationship between liver fat and IR they get. this is because there is possibility of log transformation changing the directions of correlation relationship besides the fact that it transforms non-normally distributed data into a more normal distribution.

Response

No change in directionality was observed when we used chi-square test over log-transformed values. 

Reviewer #6: While you have tried to address all the comments, the comments from the editor haven’t been sufficiently addressed. There is still some confusion between the conclusion made and what exactly was measured. You conclude that it is the liver fat that shows association and not visceral adipose tissue, whereas the visceral adipose tissue also include liver fat. Additionally, you also include metabolic syndrome in your title, discussion and conclusion while the methodology doesn’t describe or show how patients with metabolic syndrome were selected and recruited. Furthermore, your title is too conclusive while not clearly supported by data. You would rather have written “Abdominal fat depots and their association with insulin resistance in patients with type 2 diabetes” and have the freedom to describe your findings rather than having a very conclusive title. The small sample size is also hindrance for such a conclusive title.

Response

Following updates have been made to resolve the concerns of reviewers: 

1. The title of manuscript has been changed to ‘Abdominal fat depots and their association with insulin resistance in patients with type 2 diabetes’ instead of earlier conclusive title and also justifying the low sample size.

2. Methodology to assign Metabolic Syndrome to individual has been included in Material and Method section. 

3. The description and terms used for fat depots are transformed to common terms 

4. In-place of visceral fat, we used term ‘intra-peritoneal visceral adipose tissue fat mass surrounding the vital organs (IPAT-SV)’ to discriminate it from liver fat

---

## [Decision Letter · Decision Letter 4]

27 Nov 2023

Abdominal fat depots and their association with insulin resistance in patients with type 2 diabetes

PONE-D-22-13467R4

Dear Dr. Mathur,

We’re pleased to inform you that your manuscript has been judged scientifically suitable for publication and will be formally accepted for publication once it meets all outstanding technical requirements.

Kind regards,

Fredirick Lazaro mashili, MD, PhD

Academic Editor

PLOS ONE

Additional Editor Comments (optional):

All the comments have been sufficiently addressed

Reviewers' comments:

Reviewer's Responses to Questions

**Comments to the Author**

1. If the authors have adequately addressed your comments raised in a previous round of review and you feel that this manuscript is now acceptable for publication, you may indicate that here to bypass the “Comments to the Author” section, enter your conflict of interest statement in the “Confidential to Editor” section, and submit your "Accept" recommendation.

Reviewer #5: (No Response)

Reviewer #6: (No Response)

2. Is the manuscript technically sound, and do the data support the conclusions?

Reviewer #5: Yes

Reviewer #6: Partly

3. Has the statistical analysis been performed appropriately and rigorously? 

Reviewer #5: Yes

Reviewer #6: I Don't Know

4. Have the authors made all data underlying the findings in their manuscript fully available?

Reviewer #5: Yes

Reviewer #6: Yes

5. Is the manuscript presented in an intelligible fashion and written in standard English?

Reviewer #5: Yes

Reviewer #6: Yes

6. Review Comments to the Author

Reviewer #5: There are still some areas where p-values have been reported incorrectly. authors can read this article and make revisions accordingly. https://doi.org/10.1136%2Fbmjebm-2019-111264 . This article elaborates on how p values should be reported.

Reviewer #6: While you have tried to address some coments, the metabolic syndrome sub-group is still very confusing. Were those with metabolic syndrome among those with diabetes, or some were in the no diabetes group? This should have been clearly described on the study subject section of the methodology. The analysis and discussion should also support the conclusion that in individual suffering from T2D and MetS, IR shows a trend towards positive and borderline negative correlation with liver fat and SAT fatvmass respectivelly. Is it in those with MetS and T2D combined, or separatelly? This is completelly not clear.

7. PLOS authors have the option to publish the peer review history of their article (what does this mean?). If published, this will include your full peer review and any attached files.

Reviewer #5: No

Reviewer #6: **Yes: **Fredirick mashili

---

## [Editor Report · Acceptance letter]

30 Nov 2023

PONE-D-22-13467R4 

Abdominal fat depots and their association with insulin resistance in patients with type 2 diabetes 

Dear Dr. Mathur:

I'm pleased to inform you that your manuscript has been deemed suitable for publication in PLOS ONE. Congratulations! Your manuscript is now with our production department. 

Kind regards, 

on behalf of

Dr Fredirick Lazaro mashili 

Academic Editor

PLOS ONE